# The Effect of Childhood Poly-Victimization on Adulthood Aggression: The Mediating Role of Different Impulsivity Traits

**DOI:** 10.3390/bs14020100

**Published:** 2024-01-28

**Authors:** Demi C. Bartelen, Stefan Bogaerts, Marija Janković

**Affiliations:** 1Department of Developmental Psychology, Tilburg University, 5037 AB Tilburg, The Netherlandss.bogaerts@tilburguniversity.edu (S.B.); 2Fivoor Science and Treatment Innovation (FARID), 3015 CN Rotterdam, The Netherlands

**Keywords:** childhood traumatic experiences, aggression, impulsivity traits, poly-victimization

## Abstract

This study investigated the effect of the poly-victimization pattern of traumatic childhood experiences on aggression via the impulsivity traits positive urgency, negative urgency, lack of perseverance, lack of premeditation, and sensation-seeking in 102 poly-victims of childhood trauma (71.57% were females; *M*_age_ = 35.76; *SD*_age_ = 15.91). Analyses with poly-victimization as an independent variable, impulsivity traits as parallel mediators, (1) reactive aggression or (2) proactive aggression as dependent variables, and gender as a covariate revealed that the poly-victimization did not have a direct or indirect effect on reactive or proactive aggression, nor did it have an effect on any of the impulsivity traits. Moreover, lack of premeditation had a positive direct effect on reactive aggression, while gender was a significant covariate in both models, with males reporting more aggression than females. Findings suggest that the poly-victimization does not influence impulsivity traits and aggression in adulthood. However, in males, the poly-victimization had a positive and moderate correlation with reactive aggression and negative urgency, while these correlations were absent in females. This finding implies that males are more vulnerable to the adverse effects of childhood poly-victimization than females.

## 1. Introduction

Aggression poses a major societal concern due to its high prevalence and the substantial costs it incurs [1]. In the United States, more than 20% of college students engage in various forms of aggressive behavior [2]. Aggression, defined as intentional actions that harm others, can be categorized into reactive or proactive forms [3]. Reactive aggression is impulsive and emotional, triggered by perceived provocation or threats, while proactive aggression lacks emotional attachment and is motivated by goals unrelated to harming others [4]. The negative consequences of aggression affect both its victims, resulting in physical or psychological harm, and its perpetrators, who may face incarceration or other adverse outcomes. The General Aggression Model has become a widely adopted theoretical framework for understanding these mechanisms and designing effective interventions [5]. Efforts to uncover the etiological factors of aggression must consider personal and environmental factors, as aggression is a multifaceted problem driven by person-environment interactions [5].

Considering the environmental factors contributing to aggression, numerous studies have consistently linked adverse childhood experiences to aggressive behavior in both cross-sectional and longitudinal research [6,7]. Adverse childhood experiences are stressful or traumatic events that occur during childhood and can take various forms, including physical and emotional neglect, as well as physical, emotional, and sexual abuse [8,9]. In addition to childhood maltreatment, there are various other forms of adverse childhood experiences, such as household dysfunction, the incarceration of a significant other, or the separation of parents, all of which negatively impact the lives of victims in different ways. The current study focuses specifically on childhood maltreatment and its various forms. The scientific literature strongly linked traumatic childhood experiences with multiple forms of poor life outcomes, such as psychopathology, delinquency, and, as stated above, aggression [10,11].

Traumatic childhood experiences are known for their tendency to cluster together, exerting a more pronounced impact when multiple forms of adverse events are encountered [8]. For example, Feng et al. [12] found cumulative effects of adverse childhood experiences on the health of primary school children, and Roos et al. [13] noted an increased risk of adult incarceration with co-occurring adverse childhood experiences. Poly-victims, those experiencing various adverse events in childhood, faced a higher risk of poor life outcomes and aggressive behaviors due to a dose-response relationship. The systematic review by Cooke et al. [14] demonstrated that the quantity of traumatic events in childhood predicted poor life outcomes more reliably than the specific forms. Hence, this study focused on individuals exhibiting a poly-victimization pattern of childhood trauma, meaning they have experienced at least two of the five measured forms of childhood maltreatment (physical abuse, sexual abuse, emotional abuse, physical neglect, and emotional neglect).

In line with Social Learning Theory [15], poly-victims of childhood trauma may acquire aggressive behavior by observing and imitating others rewarded for this behavior. However, the General Aggression Model [5] underscored the importance of personal factors in aggression development. One such personal factor robustly linked to both traumatic childhood experiences and aggression is impulsivity [14,16]. Impulsivity, which is defined by the Diagnostic and Statistical Manual of Mental Disorders as “acting on the spur of the moment in response to immediate stimuli or acting on a momentary basis without a plan or consideration of outcomes as well as difficulty establishing and following plans,” plays a pivotal role in understanding these relationships [17] (p. 764).

Experiencing traumatic events in childhood can disrupt neurocognitive development, increasing the risk of inhibitory control deficits and impulsive behavior in adulthood [18,19]. Therefore, (poly-)victims of childhood trauma are particularly susceptible to developing impulsivity, which is related to behavioral regulation impairments and aggression [20,21,22]. For example, Perez et al. [23] found that traumatic childhood experiences predict impulsivity and aggression, with these factors partially mediating the relationship between traumatic childhood experiences and violent delinquency. Fanning et al. [24] found that experiencing physical abuse impacts the development of intermittent explosive disorder in adulthood, characterized by impulsive aggression mediated by impulsivity. Brown et al. [25] concluded that impulsivity played an important role in the relationship between emotional abuse during childhood and substance use in adulthood. Sequentially, substance use, in turn, has been consistently associated with aggression and violence across different populations, according to the meta-meta-analysis of Duke et al. [26].

The shortcoming of the aforementioned studies lies in treating impulsivity as a unidimensional construct, lacking differentiation between various impulsivity traits. This approach has resulted in inconsistencies in the conceptualization of impulsivity. Recognizing impulsivity as a multifaceted construct, subdivided into distinct traits [27,28,29], can unveil more precise associations with (the poly-victimization pattern of) traumatic childhood experiences and (reactive and proactive) aggression. These insights can be highly relevant for intervention practices. Also, considering impulsivity as a multidimensional construct in research may lead to less confusion regarding the conceptualization of impulsivity.

Whiteside and Lynam [29] created the UPPS model of impulsivity to elucidate the multifaceted nature of the personality trait impulsivity and to provide clear conceptualizations for every impulsivity aspect. Four distinct aspects were found and labelled as urgency, (lack of) premeditation, (lack of) perseverance, and sensation-seeking, hence the name UPPS. Later research by Cyders and Smith [27,28] showed the importance of further differentiating urgency into positive and negative urgency. This expansion resulted in the revised UPPS-P model, which comprised five distinct impulsivity traits [27]. Urgency reflects the tendency to engage in regrettable or impulsive actions driven by intense emotions, either positive or negative (positive urgency versus negative urgency). These are emotion-related impulsivity traits. In contrast, the remaining three impulsivity traits are non-emotion-related impulsivity traits. Lack of premeditation denotes the inclination to act without thoughtful consideration of consequences. Lack of perseverance refers to an inability to maintain focus on challenging or boring tasks. Finally, sensation-seeking refers to openness to trying new experiences and a trend to enjoy and pursue exciting activities.

Several studies explored the distinct relationships between various impulsivity traits and traumatic childhood experiences. Most studies have consistently demonstrated a strong connection between negative urgency and adverse experiences in childhood [30,31]. Focussing on poly-victims of childhood adversity, Shin et al. [32] contended that these poly-victims have higher levels of negative urgency compared to those with low or exclusively emotional traumatic childhood experiences. However, Shin et al. [32] did not find significant relations between poly-victimization and impulsivity traits, including lack of premeditation, lack of perseverance, and sensation-seeking in a community sample. Different impulsivity traits also exhibit distinct associations with reactive and proactive aggression. Hecht and Latzman [20] found that reactive aggression is notably characterized by negative urgency, followed by low perseverance, high premeditation, and low positive urgency. In contrast, proactive aggression is mainly characterized by positive urgency and, to a lesser extent, by high premeditation. Consistent with Hecht and Latzman [20], Bresin’s [16] meta-analysis concluded that negative urgency is predominantly linked to reactive aggression, while positive urgency aligns more with proactive aggression. However, Bresin [16] found that reactive aggression is also, to a lesser extent, positively correlated with positive urgency, while proactive aggression is also positively correlated with lack of perseverance and lack of premeditation. Overall, Bresin [16] argued that negative urgency, positive urgency, and lack of premeditation have small-to-medium correlations with aggression, whereas lack of perseverance and sensation-seeking only have small correlations with aggression.

Considering traumatic childhood experiences, distinct impulsivity traits, and reactive and proactive aggression collectively, Madole et al. [21] found that childhood adversity and emotion-related impulsivity traits exerted direct and unique effects on aggressive behavior. They also identified an indirect effect of childhood adversity on aggression through the pervasive influence of feelings, a facet of impulsivity that includes negative urgency. In contrast, Richey et al. [33] found that traumatic childhood experiences are associated with reactive but not proactive aggression. These results altogether suggested that poly-victims of childhood adversity predominantly exhibit impulsive behavioral control during negative emotional arousal, aligning more closely with reactive aggression than proactive aggression. Nevertheless, according to the Social Learning Theory [15], poly-victims may also express proactive aggression if they have learned from their perpetrator(s) that aggression can be used to achieve secondary goals.

While research has explored the specific associations between poly-victims of childhood adversity, distinct impulsivity traits, and reactive and proactive aggression, a comprehensive understanding of the underlying relationships still remains incomplete. This study attempts to clarify the nomological network between the poly-victimization pattern of childhood trauma, the UPPS-P impulsivity traits, and two subtypes of aggression to gain a better understanding of the complex relationships between the constructs. It is decided to divide aggression into reactive and proactive aggression based on a two-factor model acknowledging their distinct dimensions and psychosocial correlates [34,35]. Furthermore, gender is included as a controlling variable, given consistent findings that females experienced more traumatic childhood experiences while males exhibited higher levels of aggressive behavior [14,22]. Results from this study may inform targeted interventions to reduce both reactive and proactive aggression, minimizing societal costs by addressing specific impulsivity traits [4,36].

To achieve the outlined objectives, our research question is as follows: “To what extent do five different impulsivity traits mediate the relationship between poly-victimization patterns of childhood trauma and reactive and proactive aggression within a community sample?” It is firstly hypothesized that poly-victimization directly influences all five impulsivity traits. It is expected that negative urgency will be the strongest influenced impulsivity trait. Although Shin et al. [32] did not find correlations between poly-victimization and traits, such as lack of premeditation, lack of perseverance, and sensation-seeking, we derived our hypotheses from the Social Learning Theory [15]. Secondly, it is hypothesized that all impulsivity traits will have direct positive effects on reactive and proactive aggression. Negative urgency will have the strongest effect on reactive aggression, while positive urgency will have the strongest effect on proactive aggression. For the remaining three impulsivity traits, we abstained from forming specific hypotheses about their influence on reactive and proactive aggression due to mixed research findings [16,20]. Thirdly, we contemplate that poly-victimization will have direct positive effects on reactive and proactive aggression, with reactive aggression expected to be more strongly affected by poly-victimization compared to proactive aggression. Although Richey et al. [33] did not find an effect of childhood adversity on proactive aggression, it is hypothesized that poly-victims of childhood trauma can become proactive aggressive, aligning with the principles of the Social Learning Theory [15]. Lastly, it is hypothesized that poly-victimization indirectly affects reactive and proactive aggression via its impact on the five impulsivity traits. More distinctively, it is expected that negative urgency will be the strongest mediator in the association between poly-victimization and reactive aggression, while positive urgency will be the strongest mediator in the association between poly-victimization and proactive aggression. All hypotheses are derived considering all the abovementioned literature.

## 2. Materials and Methods

### 2.1. Participants

The sample included 102 participants, of which 28.4% were males and 71.6% were females, with a mean age of 35.76 years (*SD* = 15.91, range = 18–77). Most of the participants were Caucasian (83.3%) and highly educated (67.7%). For a complete overview of sample characteristics, including gender differences, see Table 1. No significant gender differences were observed in the demographic data. Participants were recruited from the general Dutch population by master’s level students from Tilburg University through Qualtrics, an online survey tool. These students distributed the survey among their friends and acquaintances through various social media platforms (e.g., Twitter, Facebook, Instagram, and LinkedIn). An a priori power analysis was done using G*Power 3 [37] to calculate the sample size. An F-test for linear multiple regression with a fixed model and R2 deviation from zero revealed that when using seven predictors, there were 103 participants needed to get a power of 80% with a significance level of α = 0.05 and a medium effect size of *f*^2^ = 0.15.

### 2.2. Measures

#### 2.2.1. Poly-Victimization Pattern of Childhood Trauma

The poly-victimization pattern of childhood trauma was measured retrospectively with the Dutch and English versions of the Child Trauma Questionnaire-Short Form (CTQ-SF) [38]. The CTQ-SF comprises 28 self-report items, of which 25 items measure five dimensions of childhood maltreatment, and three items measure denial/minimization. The denial/minimization scale was not used in this study. The five dimensions include physical abuse (e.g., “People in my family hit me so hard that it left me with bruises or marks”), sexual abuse (e.g., “Someone tried to touch me in a sexual way or tried to make me touch them”), emotional abuse (e.g., “I thought that my parents wished that I had never been born”), physical neglect (e.g., “I had to wear dirty clothes”), and emotional neglect (e.g., “I felt loved (This is a reverse-coded item.)”). Each of the childhood maltreatment scales consisted of five items on a Likert scale with a range of 1 (“never true”) through 5 (“very often true”). After reverse coding items 2, 5, 7, 13, 19, 26, and 28, a higher score on the summed childhood maltreatment (sub)scale(s) indicated more severe childhood maltreatment. The subscales for each childhood maltreatment subscale ranged from 5 to 25, while the total childhood maltreatment scale ranged from 25 to 125. Since this study focused on the poly-victimization pattern of childhood trauma, only participants with low to severe scores in at least two of the five childhood maltreatment dimensions were included in this study, using the cut-off scores from Bernstein and Fink [39]: at least 9 for physical abuse, 7 for sexual abuse, 12 for emotional abuse, 9 for physical neglect, and 14 for emotional neglect. Subsequently, the study computed a total score for the poly-victimization pattern after filtering for poly-victims. Thombs et al. [40] validated the Dutch CTQ-SF, reporting acceptable to very good Cronbach’s alphas: 0.91 for physical abuse, 0.89 for emotional abuse, 0.95 for sexual abuse, 0.63 for physical neglect, and 0.91 for emotional neglect [41]. The current study found a Cronbach’s alpha of 0.81 when using the sum score of the CTQ-SF.

#### 2.2.2. Impulsivity Traits

All five impulsivity traits were measured using the Dutch and English versions of the Short Impulsive Behavior Scale (S-UPPS-P) [42]. S-UPPS-P is a self-report questionnaire consisting of 20 items, with responses rated on a Likert scale from 1 (“strongly agree”) to 4 (“strongly disagree”). Participants rated their (dis)agreement with statements that characterized various impulsivity dimensions and thought patterns, measuring five distinct impulsivity dimensions. These dimensions were negative urgency (e.g., “When I am upset, I often act without thinking (This is a reverse-coded item.)”), positive urgency (e.g., “I tend to lose control when I am in a great mood (This is a reverse-coded item.)”), lack of premeditation (e.g., “I usually think carefully before doing anything”), lack of perseverance (e.g., “I finish what I start”), and sensation-seeking (e.g., “I quite enjoy taking risks (This is a reverse-coded item.)”). Items 3, 6, 8, 9, 10, 13, 14, 15, 16, 17, 18, and 20 required reverse coding. After reverse coding, the study generated five sum scales, with each impulsivity aspect having a possible score from 4 to 16. A higher score indicated higher levels of that particular impulsivity aspect. Cyders et al. [42] found Cronbach’s alphas for negative urgency, positive urgency, lack of premeditation, lack of perseverance, and sensation-seeking of 0.78, 0.85, 0.85, 0.79, and 0.74, respectively, in an English-speaking community sample. Cyders et al. [42] concluded that SUPPS-P had a similar factor structure to the original UPPS-P, thereby guaranteeing the validity of the instrument. In this study, Cronbach’s alphas of respectively 0.45, 0.79, 0.68, 0.53, and 0.80 were found for negative urgency, positive urgency, lack of perseverance, sensation-seeking, and lack of premeditation, indicating reliability ranging from unacceptable to acceptable [41]. It is worth noting that Cronbach’s alphas can be influenced by the number of items on a scale, with lower alphas expected for scales with fewer items [43]. This may explain the unacceptably low alphas, and therefore, inter-item correlations were checked to assess scale reliability. This study found mean inter-item correlations between 0.15 and 0.63 for the five sum scales, which aligns with acceptability criteria according to Clark and Watson [44].

#### 2.2.3. Reactive and Proactive Aggression

Reactive and proactive aggression were measured by the English Reactive and Proactive Aggression Questionnaire [45] and the Dutch version (RPQ) [46]. The RPQ is a self-report questionnaire that consists of 11 items to measure reactive aggression (e.g., “How often have you yelled at others when they annoyed you?”) and 12 items to measure proactive aggression (e.g., “How often have you yelled at others so they would do things for you?”). These items align with a two-factor model of aggression. Respondents rated the items on a Likert scale from 0 (“rarely”) to 2 (“most of the time”), with higher scores indicating higher levels of aggressive behavior. The current study created two sum scales for reactive and proactive aggression. The scoring range for reactive aggression was 0 to 22, and for proactive aggression, it was 0 to 24. Higher scores on these scales indicated more (reactive or proactive) aggression. This two-factor model of the RPQ has been successfully validated across different cultures and general populations [47]. Raine et al. [45] found Cronbach’s alphas ranging from 0.81 to 0.86 for reactive aggression and 0.84 to 0.87 for proactive aggression, depending on the sample. This study found Cronbach’s alpha’s of 0.84 and 0.74 for reactive and proactive aggression, indicating good and acceptable reliabilities [41].

### 2.3. Procedure

The Ethical Review Board of the Faculty of Social and Behavioral Sciences at Tilburg University has given ethical approval for the current study. Master’s level students recruited participants from the general Dutch population, aged 18 or older, proficient in Dutch or English. Participants received a study description, including an information letter, prior to questionnaire completion. The letter also emphasized the option to withdraw at any time without explanation. Participants who accepted the study’s objectives provided informed consent before completing the electronic questionnaires, which took about 30 min. Participation in the study was completely voluntary, with no rewards offered.

### 2.4. Statistical Analyses

The current study adopted a cross-sectional design, investigating the relationship between the poly-victimization pattern of childhood trauma as the independent variable and both reactive and proactive aggression as dependent variables. All five impulsivity traits were considered as individual mediators; gender was included as a control variable. A conceptual diagram of the analyses is provided in Figure 1.

Data analyses were conducted using IBM SPSS Statistics 26, with a significance level set at α = 0.05. The study variables were treated as continuous, while gender was categorized as a nominal, dichotomous variable. Participants who did not identify as male or female (*n* = 4) were excluded from the dataset. Moreover, participants who did not score low on at least two of the five childhood maltreatment subscales from the CTQ-SF were also excluded, resulting in the removal of 551 participants from the dataset. A summary of the demographic characteristics for both the selected and excluded sample can be found in Appendix A (Table A1). Significant differences were observed between the two samples concerning ethnicity and employment status. Although both samples consisted predominantly of Caucasians, the retained sample had a somewhat higher percentage of ethnic minorities. In addition, the excluded sample showed a higher percentage of individuals with full-time employment compared to the retained sample. However, these observed differences are not unexpected, as the study deliberately selected participants with higher scores on childhood trauma. Therefore, our sample does not accurately represent the entire population but rather reflects a subpopulation of community individuals who have experienced various forms of childhood trauma.

Outliers were traced using Cook’s Leverage and Mahalanobis distances. Participants exhibiting outliers on at least two of these three measurements were removed from the dataset, although none met this criterion, resulting in a final sample of *N* = 102. For the final sample, sum scales were generated for all study variables, and various statistical assumptions were checked. Multicollinearity was assessed through tolerance values and variation inflation factors (see Table A2 in Appendix A), Durbin-Watson statistics were computed to check the independence of error terms (see Table A3 in Appendix A), and graphical assessments were performed to confirm homoscedasticity and linearity (see Figure A1, Figure A2, Figure A3 and Figure A4 in Appendix A). These results indicated that no assumptions were violated.

Descriptive statistics were computed for all continuous variables, and Pearson’s correlation coefficients between these variables were computed after splitting the dataset by gender. Significant correlation coefficients were tested for significant differences between males and females using the Fisher r-to-z transformation by the online calculation program of Lowry [48] or the online calculation program of Lee and Preacher [49] for significant Pearson’s correlation coefficients of two dependent correlations with one study variable in common. Lastly, two mediation analyses with five parallel mediators were conducted using template 4 of PROCESS Macro version 4 in SPSS [50]. These analyses aimed to assess the direct and indirect effects of poly-victimization on reactive and proactive aggression via five different impulsivity traits. Gender was included as a covariate, with males coded as 0 and females as 1. Bootstrap procedure with *N* = 5000 resamples was used to obtain more robust estimates of indirect effects at 95% bias-corrected confidence intervals.

## 3. Results

### 3.1. Descriptive Statistics

Descriptive statistics for study variables can be found in Table 2, both for the entire sample and by gender. Overall, participants scored low on the poly-victimization scale, indicating limited childhood maltreatment. Participants scored highest on impulsivity traits, negative urgency, and sensation-seeking while scoring lowest on lack of perseverance. Moreover, participants generally had a higher score on the reactive aggression scale compared to the proactive aggression scale.

The Pearson’s correlation coefficients for the study variables are displayed in Table 3 and Table 4 for the entire sample and by gender. Effect sizes for significant correlations were interpreted following Cohen’s guidelines [51]. In the entire sample, the poly-victimization pattern of childhood trauma did not show significant correlations with other variables. Impulsivity traits exhibited moderate to large intercorrelations with each other, with moderate to large effect sizes. Moreover, there was a strong positive correlation between the two aggression subtypes. Reactive aggression correlated positively with the impulsivity trait lack of premeditation, while proactive aggression correlated positively with negative urgency, positive urgency, and lack of premeditation. All these correlations between aggression and impulsivity traits were of moderate size. Findings using Lee and Preacher’s [49] calculation program revealed that the correlation with lack of premeditation did not significantly differ between reactive and proactive aggression (*z* = −0.15, *p* = 0.882). This implies that reactive aggression did not have a stronger correlation with lack of premeditation than proactive aggression.

In the male sample, the poly-victimization did positively correlate with negative urgency and reactive aggression, which were moderate to large in size. Impulsivity traits were less intercorrelated in males, though the magnitudes of the effects were still moderate to large. Reactive and proactive aggression were strongly positively correlated in males but had no significant correlations with any of the impulsivity traits. In the female sample, poly-victimization did not show significant correlations with any other study variables. As with males, positive urgency and negative urgency showed a strong positive correlation, with a similar effect size in both genders. However, the correlation between positive and negative urgency was not significantly different in males and females (*z* = −0.66, *p* = 0.509). In females, much like in the entire sample, lack of premeditation had positive correlations with all other impulsivity traits, except for sensation-seeking, with moderate to large effect sizes. Reactive and proactive aggression showed a strong positive correlation in females, which was not significantly different from the correlation observed in males (*z* = 1.27, *p* = 0.204). In the female sample, the correlations between aggression and impulsivity followed a similar pattern to that of the entire sample. Reactive aggression did positively and moderately correlate with lack of premeditation, while proactive aggression did positively and moderately correlate with positive urgency, negative urgency, and lack of premeditation. Lastly, in the female sample, the correlation with lack of premeditation was not significantly different for reactive and proactive aggression (*z* = −0.56, *p* = 0.578).

### 3.2. Mediation Analysis with Reactive Aggression

Table 5 displays the results of the mediation analysis with reactive aggression as the dependent variable. Overall, the mediation model explained a relatively small portion of the variance in reactive aggression *F*(2, 99) = 4.72, *p* < 0.05, *R*^2^ = 0.09. The results showed that the poly-victimization pattern of childhood trauma did not exhibit any direct, indirect, or total effects on reactive aggression. Likewise, no significant effects were found between poly-victimization and any of the impulsivity traits, nor between any of the impulsivity traits and reactive aggression, except for lack of premeditation. Lack of premeditation did have a significant, positive direct effect on reactive aggression (*B* = 0.51, *SE* = 0.19, 95% CI [0.13, 0.90], *p* < 0.01) with an effect size of β = 0.30, which is considered a medium effect size. Gender served as a significant covariate in the model (*B* = −2.53, SE = 0.94, 95% CI [−4.40, −0.67], *p* < 0.01). Males scored significantly higher on reactive aggression compared to females. Gender also demonstrated significant direct effects on the impulsivity traits, specifically negative urgency (*B* = −1.17, *SE* = 0.53, 95% CI [−2.22, −0.12], *p* < 0.05) and sensation-seeking (*B* = −1.58, *SE* = 0.56, 95% CI [−2.69, −0.47], *p* < 0.01), with effect sizes of β = −0.22 and β = −0.28, indicating small to medium effect sizes. This implies males scored significantly higher than women on impulsivity traits, negative urgency, and sensation-seeking.

### 3.3. Mediation Analysis with Proactive Aggression

Table 6 presents the results of the mediation analysis focused on proactive aggression as the dependent variable. The mediation model explained a small part of the variance in proactive aggression *F*(2, 99) = 7.10, *p* < 0.01, *R*^2^ = 0.13. Similar to the model with reactive aggression, the results showed that the poly-victimization pattern of childhood trauma did not have a significant direct, indirect, or total effect on proactive aggression. Furthermore, there were no significant effects between poly-victimization and any of the impulsivity traits, nor were there significant effects between any of the impulsivity traits and proactive aggression. Gender emerged as a significant covariate within the model (*B* = −1.96, *SE* = 0.54, 95% CI [−3.04, −0.89], *p* < 0.01), with males scoring significantly higher on proactive aggression compared to females. Gender also had a significant direct effect on proactive aggression (*B* = −1.55, *SE* = 0.58, 95% CI [−2.69, −0.41], *p* < 0.01) with an effect size of β = −0.27. Again, gender had significant direct effects on the impulsivity traits, specifically negative urgency (*B* = −1.17, *SE* = 0.53, 95% CI [−2.22, −0.12], *p* < 0.05) and sensation-seeking (*B* = −1.58, *SE* = 0.56, 95% CI [−2.69, −0.47], *p* < 0.01), with effect sizes of β = −0.22 and β = −0.28, respectively. Thus, males scored significantly higher on the impulsivity traits of negative urgency and sensation-seeking compared to females. The effects of gender on proactive aggression, negative urgency, and sensation-seeking all demonstrated small to medium effect sizes.

## 4. Discussion

This study examines the effect of the poly-victimization pattern of childhood trauma on reactive and proactive aggression, considering five impulsivity traits while controlling for gender in an adult community sample. Findings reveal that poly-victimization does not directly or indirectly affect reactive or proactive aggression, nor does it directly influence impulsivity traits. However, the lack of premeditation impulsivity trait has a moderate positive direct effect on reactive aggression. Lastly, gender serves as a significant covariate in both models, with males reporting higher levels of reactive and proactive aggression than females.

Contrary to expectations, there is no evidence supporting a direct effect of poly-victimization on any of the impulsivity traits, leading to the rejection of the first hypothesis. This contradicts prior research that has shown significant effects of childhood adversity on impulsivity traits [21,23,25]. The current findings may be attributed to the generally low scores participants achieved on the CTQ-SF childhood adversity scales, suggesting that the traumatic childhood events experienced by the participants were largely non-severe and potentially insufficient to induce effects on impulsivity traits. In spite of that, among males, a positive bivariate correlation emerged between the poly-victimization and the impulsivity trait negative urgency. This finding implies gender differences in the relationship between these constructs, which will be further discussed in the following sections.

The second hypothesis is also rejected, as urgency-related impulsivity traits show no direct effects on reactive aggression, and none of the impulsivity traits have a direct effect on proactive aggression. Contrary to prior findings, such as the effect of negative urgency on reactive aggression [20], this study reveals divergent results potentially attributed to age differences. Many previous studies [20,21] examining the specific links between impulsivity traits and aggression have done so using undergraduate samples, while this study encompasses a broad age range, from 18 to 77 years. However, the underlying mechanisms leading to aggressive behavior may significantly vary across different developmental stages, and the findings from previous research on undergraduates may not readily extend to adults. For instance, consistent research indicates that adolescents are more impulsive and reckless than adults [52,53]. Consequently, impulsivity traits may exert a more substantial influence on aggressive behavior in adolescents compared to adults, aligning with the current study’s results.

Although the second hypothesis is rejected, there is an interesting finding. The impulsivity trait lack of premeditation has a moderate positive direct effect on reactive aggression. This indicates that individuals high in lack of premeditation tend to engage more in reactive aggression, which aligns with the idea that reactive aggression occurs impulsively and spontaneously. This finding supports prior studies indicating a connection between lack of premeditation and aggressive responses under provocation [54,55]. However, it contradicts most previous studies [16,20]. For instance, a meta-analysis by Bresin [16] suggested that lack of premeditation is more strongly related to proactive rather than reactive aggression. When looking at bivariate correlations, proactive aggression and lack of premeditation are as positively correlated as lack of premeditation and reactive aggression. The lack of a significant effect of lack of premeditation on proactive aggression in the mediation model might be due to shared variance with other predictors, as several impulsivity traits were moderately to largely intercorrelated. This shared variance among impulsivity traits could also account for the lack of significance observed in the mediation analyses for some other impulsivity traits.

The current finding of a positive and moderate direct effect of lack of premeditation on reactive aggression poses a challenge within the theoretical framework that distinguishes emotionally warm-blooded reactive aggression from emotionally cold-blooded proactive aggression, as lack of premeditation is considered a non-emotion-related impulsivity trait [4,27]. Hence, this suggests the need to expand the theoretical understanding of reactive aggression, as a non-emotion-related impulsivity trait seems to play a significant role in the emergence of reactive aggression. Lack of premeditation may act as a de-inhibitor, promoting immediate action in response to emotions and leading to reactive aggression when the potential consequences are not adequately considered beforehand. This finding can provide important insights for developing interventions to regulate reactive aggression in adults. Nevertheless, more research is needed to replicate this finding and to further understand the relationship between lack of premeditation and reactive aggression.

Furthermore, the poly-victimization pattern of childhood trauma does not have a direct or indirect effect on reactive or proactive aggression, leading to the rejection of the third and fourth hypotheses. These findings deviate from previous studies that identified significant effects of childhood adversity on aggression [6,7]. As previously mentioned, this discrepancy might be attributed to the generally low scores among participants on the CTQ-SF childhood adversity scales, implying that the traumatic childhood events experienced by the participants are mostly not severe in nature. This study provides a nuanced perspective on childhood trauma research. While many studies highlight childhood trauma as an enduringly detrimental experience for children with lifelong consequences, this study suggests that this is not always the case. The current study suggests that some poly-victims, especially those with relatively few and less severe traumatic childhood experiences, possess substantial resilience in coping with childhood adversity, which does not manifest in impulsive behavior and aggressive tendencies later in life.

The non-significant findings in this study regarding experienced childhood trauma may be attributed to the predominantly non-severe nature of the experienced traumatic events in childhood among participants. Another potential factor contributing to these results is the definition of the poly-victimization employed in this study. Here, poly-victimization is defined as experiencing at least two different forms of childhood trauma, reflecting the limited scoop of the instrument used in a non-clinical sample that only measures five different forms of childhood trauma. However, some previous studies use a threshold of at least four different forms of childhood adversity for defining poly-victimization [56]. This operationalization may lack diversity as the CTQ-SF measures only five types of childhood adversity, excluding others like parental divorce, life-threatening accidents, or school bullying victimization.

While acknowledging the potential for falsification of a priori hypotheses, the bivariate zero-order correlations in this study offer empirical support for seeing impulsivity as a heterogeneous construct. This perspective suggests that impulsivity comprises lower-order facets with distinct correlations and predictive validities, particularly concerning aggression. Likewise, the study found empirical evidence that reactive and proactive aggression, though strongly related, are distinguishable and may stem from different underlying mechanisms. Gender differences in the examined constructs are also evident in the results, which will be extensively discussed in the following paragraphs.

Although no a priori hypotheses were formulated for gender effects, existing research consistently indicates higher aggression levels in males compared to females [22], a trend supported by the current study’s empirical evidence of increased (self-reported) aggression in males. Aligning with previous findings on gender differences in impulsivity traits, the study supports the observation of heightened sensation-seeking in adult males. This finding is consistent with studies conducted by Argyriou et al. [57], Cyders [58], and Cross et al. [59]. However, this study also found that negative urgency is higher in males compared to females, contrary to the prevailing literature suggesting females typically score higher on negative urgency [58]. This incongruity offers valuable insights for advancing the understanding of gender differences in impulsivity traits.

The bivariate zero-order correlations reveal gender-based distinctions in the interplay between poly-victimization, impulsivity traits, and aggression. Notably, these differences may stem from variations in traditional gender roles and socialisation processes. In males, poly-victimization shows a positive and moderate correlation with both reactive aggression and negative urgency, whereas these correlations are absent in females. This suggests that males may face a heightened risk of adverse effects associated with childhood poly-victimization compared to females. Presumably, the underlying third variables influencing these correlations, such as the expression of emotions or the assertion of power, likely differ between gender due to traditional gender roles [52,60]. These variables could act as risk or protective factors against the consequences of childhood adversity. Given the medium to large effect sizes in the bivariate zero-order correlations, further investigation is both theoretically and practically pertinent. Investigating specific links within these correlations may elucidate potential gender-related differences in risk factors associated with adverse life outcomes. Understanding these differences could inform targeted intervention strategies, enhancing effectiveness in mitigating negative outcomes. However, the proposed gender differences may also result from random sampling fluctuations.

To ascertain the validity of the proposed gender differences, future studies should independently retest the models for males and females. Also, exploring additional variables contributing to observed correlations, such as personality, cognition, or environmental factors [61,62] is recommended. Conducting the study in a forensic setting, characterized by elevated levels of adverse childhood experiences, aggression, and relevant variables [22,63], is also important. For subsequent studies on poly-victimization, achieving consensus on the definition is imperative. Investigating individuals with diverse numbers and severities of childhood adversity can pinpoint when these experiences predominantly lead to adverse effects later in life. Lastly, probing specific connections between childhood trauma, impulsivity traits, and aggression across developmental stages may unveil nuanced mechanisms, potentially guiding tailored intervention strategies.

The current study has some limitations. Firstly, relying on self-report questionnaires introduces potential shared method variance and biased results. The retrospective nature of childhood trauma measurement, susceptible to recall inaccuracies, emphasizes the need for using multiple data sources and collection methods. Secondly, the cross-sectional design employed hinders establishing causal relationships among the poly-victimization pattern of childhood trauma, impulsivity traits, and (reactive and proactive) aggression. Longitudinal studies are necessary to unravel causality and directionality, examining the stability of impulsivity traits and (reactive and proactive) aggression across different life stages. Thirdly, based on the power analysis, a sample size of 103 participants was needed to test the study hypotheses. Given that the current study encompassed 102 individuals, it was slightly underpowered. Therefore, the study’s slightly underpowering increases the risk of overlooking significant effects present in the community. Replicating the study with a larger sample size could clarify whether certain effects were missed due to insufficient statistical power. Lastly, the sampling procedure employed in this study may induce bias and comprise the representativeness of the sample. This is because the recruitment relied on the personal networks of the recruiters and the preferences of the initial contacts. Given these limitations, caution is advised in interpreting results, and generalizing findings to the wider community should await further research on this topic.

In conclusion, this study shows that the poly-victimization pattern of childhood trauma does not have a direct or indirect effect on reactive or proactive aggression via distinct impulsivity traits. Notably, a noteworthy finding is the moderate direct effect of lack of premeditation on adult reactive aggression, holding both theoretical and practical relevance. Also, this study highlights gender differences in the mean levels of impulsivity traits and their relationships with the examined constructs, suggesting potential divergent pathways to aggressive behavior. Nevertheless, to comprehensively understand the aetiology of aggression in adulthood and mitigate its negative consequences, it is important to explore additional contributing factors. Preferably, this research will be conducted for males and females separately.

## Figures and Tables

**Figure 1 behavsci-14-00100-f001:**
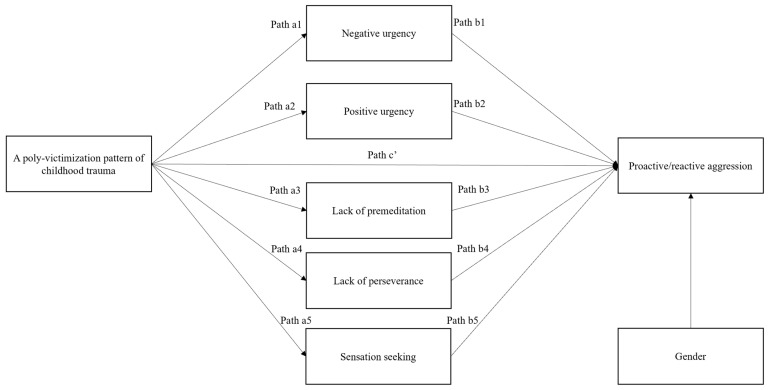
Conceptual diagram for mediation analyses. Gender is coded as a dummy variable with level 0 = male and level 1 = female.

**Table 1 behavsci-14-00100-t001:** Demographic Characteristics.

	Entire Sample (*n* = 102)	Males (*n* = 29)	Females (*n* = 73)	Test Statistic	*p*
Variable	*M* (*SD*)/*n* (%)				
Age (in years)	35.76 (15.91)	31.93 (14.49)	37.29 (16.29)	*F*(1, 100) = 2.39	0.13
Ethnicity				χ^2^(4, *N* = 102) = 7.97	0.09
Caucasian	85 (83.3%)	24 (82.8%)	61 (83.6%)		
Hispanic/Latino	0 (0.0%)	0 (0.0%)	0 (0.0%)		
African American	2 (2.0%)	2 (3.4%)	0 (0.0%)		
Asian	2 (2.0%)	1 (3.4%)	1 (1.4%)		
Prefer not to say	2 (2.0%)	1 (3.4%)	1 (1.4%)		
Other	11 (10.8%)	1 (3.4%)	10 (13.7%)		
Level of education				χ^2^(4, *N* = 102) = 4.19	0.38
No formal education	1 (1.0%)	1 (3.4%)	0 (0.0%)		
Primary school	2 (2.0%)	1 (3.4%)	1 (1.4%)		
High school	22 (21.6%)	8 (27.6%)	14 (19.2%)		
College/University	69 (67.6%)	17 (58.6%)	52 (71.2%)		
Graduate school	8 (7.8%)	2 (6.9%)	6 (8.2%)		
Marital status				χ^2^(4, *N* = 102) = 5.26	0.26
Single	46 (45.1%)	14 (48.3%)	32 (43.8%)		
In a relationship	31 (30.4%)	12 (41.4%)	19 (26.0%)		
Married	22 (21.6%)	3 (10.3%)	19 (26.0%)		
Divorced	2 (2.0%)	0 (0.0%)	2 (2.7%)		
Widowed	1 (1.0%)	0 (0.0%)	1 (1.4%)		
Employment status				χ^2^(6, *N* = 102) = 11.63	0.07
Full-time employment	30 (20.4%)	14 (48.3%)	16 (21.9%)		
Part-time employment	28 (27.5%)	4 (13.8%)	24 (32.9%)		
Unemployed/Looking for work	1 (1.0%)	0 (0.0%)	1 (1.4%)		
Unemployed/Not looking for work	2 (2.0%)	0 (0.0%)	2 (2.7%)		
Student	25 (24.5%)	9 (31.0%)	16 (21.9%)		
Retired	6 (5.9%)	1 (3.4%)	5 (6.8%)		
Other	10 (9.8%)	1 (3.4%)	9 (12.3%)		

Note. Test statistic refers to the test that was used to evaluate gender differences.

**Table 2 behavsci-14-00100-t002:** Descriptive Statistics of Study Variables.

	Entire Sample (*n* = 102)			Males (*n* = 29)			Females (*n* = 73)		
Variable	*M*	*SD*	Range	*M*	*SD*	Range	*M*	*SD*	Range
Poly-victimization pattern of childhood trauma	53.43	11.88	39–103	53.69	11.38	39–77	53.33	12.14	39–103
Negative urgency	10.36	2.45	4–16	11.21	2.47	6–16	10.03	2.38	4–16
Positive urgency	8.50	3.43	4–16	9.48	3.34	4–13	8.11	3.40	4–16
Lack of premeditation	7.71	2.55	4–16	8.28	2.15	4–12	7.48	2.67	4–16
Lack of perseverance	7.42	2.32	4–15	8.14	2.46	5–15	7.14	2.21	4–14
Sensation-seeking	10.63	2.62	4–16	11.76	2.47	8–16	10.18	2.66	4–16
Proactive aggression	2.38	2.60	0–12	3.79	3.20	0–12	1.82	2.10	0–9
Reactive aggression	7.93	4.43	0–19	9.76	5.06	0–19	7.21	3.97	0–17

**Table 3 behavsci-14-00100-t003:** Zero-order Pearson’s intercorrelations of study variables for the entire sample.

Variable	1	2	3	4	5	6	7	8
1. Poly-victimization pattern of childhood trauma	-							
2. Negative urgency	0.072	-						
3. Positive urgency	0.098	0.504 **	-					
4. Lack of premeditation	0.064	0.333 **	0.331 **	-				
5. Lack of perseverance	0.022	0.049	0.018	0.410 **	-			
6. Sensation-seeking	−0.025	0.043	0.254 **	−0.025	−0.125	-		
7. Proactive aggression	0.092	0.276 **	0.276 **	0.287 **	0.168	0.094	-	
8. Reactive aggression	0.141	0.142	0.019	0.300 **	0.168	0.104	0.573 **	-

** *p* < 0.01, two-tailed.

**Table 4 behavsci-14-00100-t004:** Zero-order Pearson’s intercorrelations of study variables disaggregated by gender.

Variable	1	2	3	4	5	6	7	8
1. Poly-victimization pattern of childhood trauma	-	−0.042	0.144	−0.009	0.075	−0.055	−0.007	−0.001
2. Negative urgency	0.368 *	-	0.518 **	0.300 **	0.015	0.026	0.313 **	0.079
3. Positive urgency	−0.034	0.398 *	-	0.372 **	0.009	0.158	0.268 *	−0.003
4. Lack of premeditation	0.307	0.358	0.130	-	0.464 **	−0.073	0.385 **	0.321 **
5. Lack of perseverance	−0.118	−0.011	−0.082	0.208	-	−0.063	0.059	0.160
6. Sensation-seeking	0.054	−0.148	0.405 *	−0.046	−0.548 **	-	−0.014	0.045
7. Proactive aggression	0.142	0.688	0.183	0.024	0.324	0.028	-	0.448 **
8. Reactive aggression	0.456 *	0.116	−0.086	0.180	0.060	0.014	0.649 **	-

Note. The results for the female sample (*n* = 73) are shown above the diagonal. The results for the male sample (*n* = 29) are shown below the diagonal. * *p* < 0.05. ** *p* < 0.01, two-tailed.

**Table 5 behavsci-14-00100-t005:** Mediation model examining effects between poly-victimization, distinct impulsivity traits, and reactive aggression.

Independent Variable (IV)	Dependent Variable (DV)	Mediating Variable (M)	Path a		Path b		Path c		Path c′		Path a*b	
			*B*	*SE*	*B*	*SE*	*B*	*SE*	*B*	*SE*	*B*	95% CI
Poly-Victimization Pattern of Childhood Trauma	Reactive aggression						0.05	0.04	0.05	0.04	0.00	[−0.03, 0.04]
		Positive Urgency	0.03	0.03	−0.26	0.15					−0.01	[−0.03, 0.01]
		Negative Urgency	0.01	0.02	0.15	0.20					0.00	[−0.01, 0.02]
		Premeditation (lack of)	0.01	0.02	0.51 **	0.19					0.01	[−0.02, 0.04]
		Perseverance (lack of)	0.00	0.02	0.03	0.20					0.00	[−0.01, 0.01]
		Sensation-seeking	−0.01	0.02	0.18	0.17					−0.00	[−0.01, 0.01]

Note. a = effect of IV on M; b = effect of M on DV; c = total effect of IV on DV; c′ = direct effect of IV on DV; a*b= indirect effect of IV on DV through M; ** *p* < 0.01.

**Table 6 behavsci-14-00100-t006:** Mediation model examining effects between poly-victimization, distinct impulsivity traits, and proactive aggression.

Independent Variable (IV)	Dependent Variable (DV)	Mediating Variable (M)	Path a		Path b		Path c		Path c′		Path a*b	
			*B*	*SE*	*B*	*SE*	*B*	*SE*	*B*	*SE*	*B*	95% CI
Poly-victimization pattern of childhood trauma	Proactive aggression						0.02	0.02	0.01	0.02	0.01	[−0.01, 0.03]
		Positive urgency	0.03	0.03	0.09	0.09					0.00	[−0.00, 0.01]
		Negative urgency	0.01	0.02	0.10	0.12					0.00	[−0.00, 0.01]
		Premeditation (lack of)	0.01	0.02	0.16	0.58					0.00	[−0.01, 0.02]
		Perseverance (lack of)	0.00	0.02	0.05	0.12					0.00	[−0.00, 0.01]
		Sensation-seeking	−0.01	0.02	−0.00	0.10					0.00	[−0.00, 0.00]

Note. a = effect of IV on M; b = effect of M on DV; c = total effect of IV on DV; c′ = direct effect of IV on DV; a*b= indirect effect of IV on DV through M. Analysis controlled for gender.

## Data Availability

The data presented in this study are available on request from the corresponding author.

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
