# Peer review of "The Effect of Childhood Poly-Victimization on Adulthood Aggression: The Mediating Role of Different Impulsivity Traits"

_behavsci, 2024, doi:10.3390/bs14020100_

Round 1

Reviewer 1 Report

Comments and Suggestions for Authors

The article discusses a relevant social issue related to the pattern of polyvictimization, seeking to investigate the effect of adverse childhood experiences (ACEs) on aggression through impulsivity traits. The article reveals scientific quality, clearly exposing the problem, and supporting some of the most relevant findings from a theoretical and empirical point of view. The article highlights the central concepts of this study, the reading models known and to be tested, proposing the consideration of impulsivity as a multifaceted construct, which is the basis of the research. The gap or argument for the study has been identified, which has led to the identification of the central research question and the definition of four well-formulated and theoretically supported research hypotheses. Methodologically, the study is well-constructed and technically sound. The sample and the sampling process are well portrayed; the variables are clearly defined and detailed. The instruments used are well characterized and the use of Dutch or English language versions is justified. The collection procedures confirm the adjusted nature of the research, as well as the statistical analysis and treatment have been explained (even schematically) and are correct. In terms of results, these are presented openly and clearly, discussing both the expected and the disproved hypotheses, as well as theoretical reflection, giving clues for possible forms of research in future studies, and highlighting the limitations that the authors' study may have had. In formal terms, there is uniformity in the use of an editing standard, the authors referenced in the text are listed (and vice versa) and the length of the article is adequate. 

Author Response

Dear reviewer,

Thank you for taking the time to review our manuscript and providing us with valuable feedback. Considering the feedback we received from you and the other two reviewers, we have revised the manuscript slightly to improve its quality. All changes are marked with track changes in the revised version. We hope this has further strengthened our work.

Marija Jankovic

Reviewer 2 Report

Comments and Suggestions for Authors

The article presented is well structured and focuses on an extremely relevant topic. The theoretical foundation is quite complete, exhaustive and clearly presented. The method is also fully described, with the measurements used being described in as much detail as necessary and presenting a good description of the sample under study. The choices regarding statistical procedures were appropriate and meet the objectives of the study. The results are presented in a concise and precise way, leaving no room for doubts about them. The discussion is of excellent quality, establishing a clear link between previous studies and the results obtained, and advancing in a simple way possible explanations for the difference between results. The limitations were correctly identified, and the suggestions for future studies are interesting and aligned with the identified limitations. 

Author Response

(The authors gave the same response as above.)

Reviewer 3 Report

Comments and Suggestions for Authors

First, thank you for the interesting paper. Please see enclosed my comments and suggestions:

-       Please provide your rationale for choosing “quota sampling among acquaintances” (Line 177) and if you control in your analysis for clustering of your sample by their recruitment person and explain your approach.

-       You have mentioned that the scale consists of 28 items with scores 1-5 (lines 186, 192-193), but the potential total score would be 25-125 (Line 197), please explain if you dropped some items from the scale.

-       Have you measured any family dysfunctionality, a traditional part of ACEs, or you have focused on only part of the child maltreatment (CM) part of it? If you focused on only CM, I’d suggest adjusting your language accordingly. 

-       You have mentioned that you excluded 551 participants from the original sample (Line 273), please provide a detailed comparison of the original and final samples’ demographics and comparison to make sure that the final sample mimics the original quota sampling characteristics. 

-       You mentioned that you needed a minimum of 103 participants for your analysis (Lines 178 - 181), but your final sample was 102, please explain how it affected the power of your analysis.

-       Please clarify if you have observed any statistical differences by gender in your descriptive analysis (Table 1) and provide N of subjects for each gender group.

-       Please provide more details for your rationale for removing 551 participants who “did not score 271 low on at least two of the five childhood maltreatment subscales from the CTQ-SF” (Line 271-273) as it limits your sample to only individuals of multiple CM experiences. 

Author Response

Dear reviewer,

Thank you for your insightful comments on our manuscript. They helped us strengthen our paper. Below we provide the point-by-point responses. All changes are marked with track changes in the revised version.

Marija Jankovic

Comments and Suggestions for Authors

First, thank you for the interesting paper. Please see enclosed my comments and suggestions:

- Please provide your rationale for choosing “quota sampling among acquaintances” (Line 177) and if you control in your analysis for clustering of your sample by their recruitment person and explain your approach.

Response: Thank you for bringing this to our attention. By reading your comment, we realized that the term “quota sampling” was not appropriate for describing our sampling methodology. The data collection was done by master’s students who distributed the survey via their social media platforms and among friends and acquaintances. Although recruiting participants via social media platforms or social networks by using 'snowballing' and word-of-mouth methods has proven effective, we cannot assert that our sample was completely random. Yet, we believe it was sufficiently diverse and reflective of the population. We deleted the term “quota sampling” to avoid any confusion and explicitly addressed the limitations of our sampling methodology in the section on limitations.

See lines 188-194 and 577-580.

- You have mentioned that the scale consists of 28 items with scores 1-5 (lines 186, 192-193), but the potential total score would be 25-125 (Line 197), please explain if you dropped some items from the scale.

Response: Thank you for your comment. Indeed, three items measuring denial/ minimization were dropped. We clarified this in the revised version of the paper. The CTQ-SF consists of 28 items of which 25 items measure five dimensions of childhood maltreatment and three items measure denial/minimization. The denial/minimization scale was not used in this study.

See lines 206-208.

- Have you measured any family dysfunctionality, a traditional part of ACEs, or you have focused on only part of the child maltreatment (CM) part of it? If you focused on only CM, I’d suggest adjusting your language accordingly.

Response: Thank you for your comment. Unfortunately, we only measured childhood trauma. We clarified this in the manuscript (See lines 49-50) and adjusted the language accordingly.

- You have mentioned that you excluded 551 participants from the original sample (Line 273), please provide a detailed comparison of the original and final samples’ demographics and comparison to make sure that the final sample mimics the original quota sampling characteristics.

Response: Thank you for your valuable input. We have incorporated your suggestion and included details the of demographic characteristics of both samples in the appendix (See Table A1). We conducted tests to identify significant differences in demographic data between the two samples. The results showed that these two samples significantly differed in ethnicity and employment status. Although both samples consisted predominantly of Caucasians, the retained sample had a somewhat higher percentage of ethnic minorities. In addition, the excluded sample exhibited a higher percentage of individuals with full-time employment compared to the retained sample. However, these observed differences are anticipated, as the study initially selected participants with higher scores on childhood trauma. Therefore, our sample does not aim to replicate the population as a whole but rather reflects a subpopulation of community individuals who have experienced multiple forms of childhood trauma.

We also clarified this in the text. See lines 295-305.

- You mentioned that you needed a minimum of 103 participants for your analysis (Lines 178 - 181), but your final sample was 102, please explain how it affected the power of your analysis.

Response: Thank you for your comment. We have updated the limitations section to include the acknowledgment that the study was slightly underpowered. In addition, we mentioned that this slight underpowering increases the risk of overlooking significant effects within the community. 

See lines 572-574.

- Please clarify if you have observed any statistical differences by gender in your descriptive analysis (Table 1) and provide N of subjects for each gender group.

Response: Thank you for your comment. We provided the table with a complete overview of sample characteristics including gender differences. We did not find any significant gender differences in demographic data.

See lines 188-194 and Table 1 on page 4.

- Please provide more details for your rationale for removing 551 participants who “did not score 271 low on at least two of the five childhood maltreatment subscales from the CTQ-SF” (Line 271-273) as it limits your sample to only individuals of multiple CM experiences.

Response: Thank you for your comment. Indeed, our sample was intentionally limited to only those with multiple CM experiences. This aligned with the aims of our study to test the extent to which, in this specific group of individuals (in the literature often referred to as poly-victims), these experiences predict aggression and whether impulsivity traits may serve as explanatory variables in this link. Although it would be equally valuable to investigate differences between poly-victims and those without multiple traumatic experiences, it was not a goal of our study. We focused particularly on poly-victims of childhood trauma, a group that has received limited attention in research until recently. See, for example:

  1. Feng, J.-Y.; Hsieh, Y.-P.; Hwa, H.-L.; Huang, C.-Y.; Wei, H.-S.; Shen, A.C.-T. Childhood Poly-Victimization and Children’s Health: A Nationally Representative Study. Child Abuse Negl. 2019, 91, 88–94, doi:10.1016/j.chiabu.2019.02.013.
  2. Finkelhor, D., Ormrod, R. K., & Turner, H.A. Polyvictimization and Trauma in a National Longitudinal Cohort. Psychopathol. 2007, 19, doi:10.1017/S0954579407070083.
